# Biological Synthesis of Silver Nanoparticles by *Amaryllis vittata* (L.) Herit: From Antimicrobial to Biomedical Applications

**DOI:** 10.3390/ma15165478

**Published:** 2022-08-09

**Authors:** Sehrish Asad, Natasha Anwar, Mohib Shah, Zeeshan Anwar, Muhammad Arif, Mamoona Rauf, Kazim Ali, Muddaser Shah, Waheed Murad, Ghadeer M. Albadrani, Ahmed E. Altyar, Mohamed M. Abdel-Daim

**Affiliations:** 1Department of Botany, Abdul Wali Khan University Mardan, Mardan 23200, Pakistan; 2Department of Chemistry, Abdul Wali Khan University Mardan, Mardan 23200, Pakistan; 3Department of Pharmacy, Abdul Wali Khan University Mardan, Mardan 23200, Pakistan; 4Department of Biotechnology, Abdul Wali Khan University Mardan, Mardan 23200, Pakistan; 5National Institute for Genomics and Advanced Biotechnology, NARC, Islamabad 44000, Pakistan; 6Natural and Medical Sciences Research Center, University of Nizwa, Birkat Al Mauz, P.O. Box 33, Nizwa 616, Oman; 7Department of Biology, College of Science, Princess Nourah bint Abdulrahman University, P.O. Box 84428, Riyadh 11671, Saudi Arabia; 8Department of Pharmacy Practice, Faculty of Pharmacy, King Abdulaziz University, P.O. Box 80260, Jeddah 21589, Saudi Arabia; 9Department of Pharmaceutical Sciences, Pharmacy Program, Batterjee Medical College, P.O. Box 6231, Jeddah 21442, Saudi Arabia; 10Pharmacology Department, Faculty of Veterinary Medicine, Suez Canal University, Ismailia 41522, Egypt

**Keywords:** *Amaryllis vittata* (L.) Herit, silver nanoparticles, spectroscopy, in vitro and in vivo activities, *Solanum tuberosum* micropropagation

## Abstract

The current study sought to synthesize silver nanoparticles (AgNPs) from *Amaryllis vittata* (L.) leaf and bulb extracts in order to determine their biological significance and use the toxic plants for human health benefits. The formation of silver nanoparticles was detected by a change in color from whitish to brown for bulb-AgNPs and from light green to dark brown for leaf-AgNPs. For the optimization of silver nanoparticles, various experimental physicochemical parameters such as pH, temperature, and salt were determined. UV-vis spectroscopy, Fourier transform infrared spectroscopy, X-ray dispersion spectroscopy, scanning electron microscopy, and energy dispersion spectroscopy analysis were used to characterize nanoparticles. Despite the fact that flavonoids in plant extracts were implicated in the reduction and capping procedure, the prepared nanoparticles demonstrated maximum absorbency between 400 and 500 nm. SEM analysis confirmed the preparation of monodispersed spherical crystalline particles with fcc structure. The bioinspired nanoparticles were found to show effective insecticidal activity against *Tribolium castaneum* and phytotoxic activity against *Lemna aequincotialis*. In comparison to plant extracts alone, the tested fabricated nanoparticles showed significant potential to scavenge free radicals and relieve pain. Antibacterial testing against human pathogenic strains, i.e., *Escherichia coli* and *Pseudomonas aureginosa*, and antifungal testing against *Aspergillus niger* revealed the significant potential for microbe resistance using AgNPs. As a result of the findings, the tested silver nanoparticles demonstrated promising potential for developing new and effective pharmacological and agricultural medications. Furthermore, the effects of biogenic AgNPs on an in vitro culture of *Solanum tuberosum* L. plants were investigated, and the findings indicated that bulb-AgNPs and leaf-AgNPs produced biomass and induced antioxidants via their active constituents. As a result, bulb-AgNPs and leaf-AgNPs may be recommended for use in *Solanum tuberosum* L. tissue culture for biomass fabrication and metabolic induction.

## 1. Introduction

Nanotechnology is known as the art of synthesizing particles with at least one dimension in the range of 1–100 nm, resulting in large surface-to-volume ratios. Nanotechnology is rapidly emerging in food, pharmaceutics [1], cosmetics, electronics, diagnostics, water treatment, agriculture, etc. [2,3]. The use of metallic nanoparticles has increased significantly in the diagnosis and therapy of various diseases [4]. Nanoparticles present outstanding outcomes in the field of nanotechnology and contribute to resolving the issues of the present decade. For the development of novel products that benefit the environment, pharmaceutical and food industry researchers combine biotechnology with green chemistry. Instead of a chemical approach, an alternative green approach is used for the synthesis of nanoparticles. The most important role of green chemistry is the formation of AgNPs from herbal extract because the hazardous chemicals coated on the surface of nanoparticles during chemical methods cannot be detached easily and inhibit biomedical applications [5]. Among the metallic nanoparticles, Ag is the most potent due to its biocompatibility and electronic property. AgNPs are a type of noble metallic nanomaterials with a large range of applications. AgNPs have drawn a lot of interest among the metals listed above because of their specific properties for use in pharmaceutics, irrigation, water detoxification, air filtration, textile industries, and as a catalyst in oxidation reactions [6]. AgNPs are primarily used in electronic batteries, glass and ceramic pigments, and medical instruments to treat cancer, HIV, diabetes, malaria, and tuberculosis [7]. Silver nanoparticles are also used for the detection of biological molecules and possess catalytic properties [8,9,10]. Nanoparticles are known for their feasible production of biomass and natural antioxidants through plant tissue cultures, along with their role in reducing contamination and inducing the bioactive secondary metabolites of plants [11,12,13]. It has been proven that metal nanoparticles are highly reactive toward plants: due to their small size, they can easily penetrate the cell membrane and accumulate inside the cell. Currently, plants are increasingly used in nanoparticle synthesis to avoid the limitation of fabricating nanomaterials using microorganisms. Several plant species, i.e., *Aloe vera*, *Jatropha curcas*, *Citrus aurantium* (Orange blossom), *Solanum melongena* (Eggplant), and *Trifolium resupinatum* [14], are known for their capability to reduce silver ions and fabricate silver nanoparticles. Various phyto constituents such as proteins, carbohydrates, fixed oil, fats, flavonoids, phenol, alkaloids, terpenoids, volatile oil, and polysaccharides are present in plants. These phytoconstituents act as stabilizing and reducing agents for silver nanoparticles. The synthesis and size of produced nanoparticles are mainly controlled by the active ingredient present in the plant. Therefore, the selection of plants for generating nanoparticles is very important. Due to this reason, *Amaryllis vittata* (L.) Herit., a poisonous plant, was selected to synthesize silver nanoparticles, which contain flavonoids, alkaloids, and coumarins and have power over carcinogenic, oxidation, and inflammatory effects. 

*Amaryllis vittata* belongs to the family Amaryllidaceae and is represented by a single genus Amaryllis (Figure 1). It is a small genus of flowering bulbs with two species. The bulbs are poisonous and contain lycorine which causes nausea and vomiting. It is a bulbous plant, with each bulb measuring 5–10 cm in diameter. The leaves are arranged in two rows (30–50 cm) long and (2–3 cm) broad. The leaves are produced in the autumn or early spring in warm regions depending on the onset of precipitation and eventually die in late spring. Each bulb produces 1–2 leafless, stout, and erect stems. The usual color is pink or purple with short stamens. The stigma is three-lobed and the seeds are compressed and globose. The current study aimed to carry out a single-step synthesis of AgNPs using leaf extract of *A. vittata*. The synthesized AgNPs were characterized by different techniques, such as FTIR spectroscopy, SEM, EDX, XRD, and TEM analysis. The nanoparticles’ antibacterial [15], antioxidant, phytotoxic, antifungal, insecticidal, and analgesic activities were also investigated to explore their biomedical potentiality.

## 2. Materials and Methods

### 2.1. Plant Collection and Identification 

The aerial parts of *A. vittata* were collected in July 2019 from Mardan and were identified by Dr Mohib Shah, Associate Professor of Botany, Department of Botany, Abdul Wali Khan University Mardan, and stored in the herbarium Department of Botany (AWKUM⁄Herb⁄2307) for future studies.

### 2.2. Preparation of Silver Nitrate

One millimolar AgNO_3_ solution (Merck, Darmstadt, Germany) was prepared in a 250 mL volumetric flask using distilled water.

### 2.3. Extract Preparation 

Leaf extract was prepared by adding 100 g of leaf powdered *A. vittata* in 500 mL ethanol in a beaker. The beaker was kept at room temperature for 7 days with occasional shaking. Then, the sample was filtered through filter paper. The filtrate was subjected to evaporation with the help of a rotary evaporator at 60 °C; the concentrated solution was transferred to Petri plates and placed in a water bath at 60 °C for complete drying. For the preparation of aqueous bulb extract, 25 g of chopped bulb were added to 100 mL distilled water in a flask and boiled in Kjeldhal apparatus at 50 °C for 10 min. Then, the sample was filtered through filter paper, and the filtrate was used as a reducing agent for silver nanoparticle synthesis.

All samples were placed on a magnetic stirrer at room temperature for 24 h. The appearance of brown color indicated the reduction of silver ions and the formation of nanoparticles. Color change for leaf-capped AgNPs was observed after 10 min of reaction, while for bulb-capped AgNPs, color change was noted soon after mixing the reactants. The rapid change in color for leaves and bulb extracts was observed in the ratio 1:5 as compared to all other samples of various salt to plant ratios. Maximum peak absorbency was observed with the help of Uv-vis spectroscopy.

### 2.4. Optimization

In the present work, the determination of various optimization parameters, i.e., temperature effect, pH effect, and NaCl concentration effect, were carried out upon the biofabrication of AgNPs.

### 2.5. UV-vis Spectroscopy

AgNPs were synthesized using leaf and bulb extracts of *A. vittata* as reducing agents, and AgNO_3_ solution was confirmed with the help of UV-vis spectrometry analysis. The synthesized silver nanoparticles were tested for absorbency against distilled water as a blank. UV-vis spectroscopy was carried out using a UV-1602 double beam UV-vis spectrometer with a resolution of 1nm to monitor the synthesis of silver nanoparticles. The spectral absorbency was evaluated with wavelengths ranging from 300 nm to 800 nm for AgNPs fabricated from the reaction mixture in different ratios such as 1:1, 1:2, 1:3, 1:4, 1:5, and 1:6, respectively.

### 2.6. FT-IR Spectroscopy

The silver nanoparticles were dried and mixed with KBr and pressed by a Hydraulic pellet press to prepare the sample pellet. The sample was subjected to FT-IR spectroscopy using the “PerkinElmer spectrometer FT-IR SPECTRUM ONE” (Spectra Lab Scientific Inc., Markham, ON, Canada) at Department of Chemistry, Bacha Khan University, Charsadda, Pakistan, via a resolution of 4 cm^−1^ in the range of 4000 cm^−1^–0 cm^−1^. The bulb and leaf extracts of *A. vittata* were then subjected to FT-IR spectroscopy, respectively, using the same procedure as for nanoparticles of *A. vittata* bulb and leaf extract, i.e., using the “Perkin Elmer spectrometer FT-IR SPECTRUM ONE” at a resolution of 4 cm^−1^ in the range of 4000 cm^−1^–0 cm^−1^.

### 2.7. X-ray Diffraction (XRD) Analysis

The X-ray diffraction pattern of Ag nanoparticles synthesized from bulk and leaf extract of *A. vittata* were analyzed using a JEOL JDX 3532 X-ray diffractometer (Lab tech.,Yokosuka, Japan) at the University of Peshawer. The diffraction pattern of biogenic nanoparticles was recorded by a nickel monochromator filtering the wave at a tube voltage of 40 KV and tube current of 30 mA with cu-kα radiation (λ = 1.5406 A_0_). The average diameter of nanoparticles was calculated from the line width of the maximum intensity reflection peak. The size of Ag nanoparticles was calculated through Scherer’s Equation (1).
D = Kλ/(β1/2cosθ)(1)
where “D” represents the average crystalline domain size perpendicular to reflecting planes. λ is the x-ray wavelength (λ = 1.5406A_0_), and K (0.89) is Scherer’s coefficient. Θ is Bragg’s angle, and β is the full width at half maximum (FWHM) in the radius.

### 2.8. Scanning Electron Microscopy

Silver nanoparticles prepared from the leaf and bulb extract of *A. vittata* were completely air-dried at 35 °C with the help of a vacuum drier, then a small droplet of leaf-AgNPs and bulb-AgNPs were dropped on a carbon-coated SEM grid, respectively, and allowed to dry. The samples were then subjected to silver coating with the help of a spi-module sputter coater (Auto fine coater), and morphological features were analyzed by scanning electron microscopy using FE-SEM (JSM-5910-JEOL) (Lab tech.,Yokosuka, Japan) at the University of Peshawer.

### 2.9. Energy-Dispersion Spectroscopy (EDS)

EDS assessment was applied to observe the confirmation of elemental Ag present in AgNPs. The sample was prepared by centrifugation of nanoparticles solution at 14,800 rpm for 15 min and allowed to dry at 35 °C by vacuum drier. The dry powder mass (nanoparticles) was then subjected to energy-dispersion spectroscopy analysis with the help of a thermal EDS instrument attached to an Oxford Inca 200 SEM, (Lab tech., Yokosuka, Japan) at the University of Peshawer.

### 2.10. Biological Activities

#### 2.10.1. Insecticidal Screening 

Leaf and bulb extracts and silver nanoparticles were tested for insecticidal activity against *Tribolium castaneum*. All samples were tested in Petri plates under aseptic conditions with two replicates. The filter papers were cut according to the size of Petri dishes and were placed on plates. Approximately 1.5 mg of sample were dissolved in 1.5 mL solvent (distilled water) serving as stock solution. Three replicates of sterilized Petri dishes were incubated with 100 µL, 500 µL, and 1000 µL solution over filter paper with the help of a micropipette. The plates were kept for 24 h at 25 °C to evaporate the solvent completely. After 24 h, stored insects obtained from a pharmacognosy lab were deposited on each Petri plate. After the evaporation of the solvent, *Tribolium castaneum* (30 healthy insects) were placed on each plate (test and control) with the help of a clean brush. Healthy insects of the same age and size were selected. The plates were incubated at 27 °C for 24 h with 60% relative humidity in the growth chamber. The number of living insects was counted and examined daily during incubation. The Petri plates remained there for 3 days. The number of insects/plates was counted and recorded on the third day. The result was analyzed using (SPSS) in terms of percentage. The mortality percentage was calculated with the help of the following Equation (2).
(2)$$\mathrm{Mortality}\%=\frac{100-\mathrm{No}.\;\mathrm{of}\;\mathrm{insects}\;\mathrm{alive}\;\mathrm{in}\;a\;\mathrm{test}\times 100}{\mathrm{Total}\;\mathrm{No}.\;\mathrm{of}\;\mathrm{insects}\;\mathrm{in}\;\mathrm{control}}.$$

The insecticidal activity was determined using the following method [16].

#### 2.10.2. Phytotoxic Activity

Phytotoxic activity analysis was carried out to evaluate the phytotoxic potential of silver nanoparticles synthesized using leaf and bulb extracts of *A. vittata*. The experiment was carried out in Petri dishes in three different concentrations (50 µL/mL/, 100 µL/mL, 500 µL/mL) with three replicates. The Petri dishes were autoclaved at 121 °C for 24 hrs. *Lemna aequinoctialis* was used as a test plant, with 10 plants per Petri dish. The extracts (1 mg) were dissolved in 1 mL distilled water and served as a stock solution (1000 ppm). Three concentrations, i.e., 50, 100, and 500 µL, were prepared from 1000 ppm solution. Ten plants containing rosettes of three fronds were kept on each plate, and samples were added to all plates containing *Lemna aequinoctialis*. The same protocol was followed for bulb extract, leaf-AgNPs, and bulb-AgNPs. Other plates were supplemented with L. medium, and reference growth inhibitor (Atrazine) served as a negative and positive control, respectively. All plates were kept in a growth chamber at 28 °C for seven days. The number of fronds per plate was counted and recorded on the seventh day.

Percentage growth inhibition was recorded with reference to the negative control using Equation (3).
(3)$$\mathrm{Inhibition}\%=\frac{\mathrm{No}\;\mathrm{of}\;\mathrm{fronds}\;\mathrm{in}\;\mathrm{test}\times 100}{\mathrm{No}\;\mathrm{of}\;\mathrm{fronds}\;\mathrm{in}\;\mathrm{control}}.$$

Phytotoxic activity analysis was carried out following the standard protocols of Ref. [17].

#### 2.10.3. Antioxidant Activity

A comparative antioxidant assay of silver nanoparticles and plant extracts of *A. vittata* was evaluated using DPPH (1-diphenyl-2-picrylhydrazyl) solution [18,19,20]. Nanoparticles and plant extracts were determined for their measurement of reducing potential and scavenging action on free radicals produced by DPPH solution by comparing them with a standard antioxidant drug (Ascorbic acid). All samples with three concentrations (50 µL/mL, 100 µL/mL, 500 µL/mL) and three replicates were prepared in distilled water. Next, 0.004% *w*/*v* DPPH solution was prepared in 99.9% methanol, and various concentrations of all samples were added to 2 mL fresh solution of DPPH in test tubes, respectively, so that the final volume reached 3 mL. Samples were shaken vigorously and incubated at room temperature for 60 min in the dark. For standard antioxidant drugs, ascorbic acid was used, and the same protocol was followed for the preparation of different concentrations of the standard drug. After incubation for 60 min, the absorbency of all samples was calculated at 517 nm against the blank (methanol) solution using a UV-spectrophotometer. The low absorbency value of the spectrophotometer revealed high free radical scavenging potential, while higher absorbency showed low scavenging activity. The percentage of antioxidant potential for all samples was calculated via Equation (4).
(4)$$\%\;\mathrm{antioxidant}\;\mathrm{potential}\;=\frac{\mathrm{Absorbance}\;\mathrm{of}\;\mathrm{control}\left(\mathrm{nm}\right)-\mathrm{absorbance}\;\mathrm{of}\;\mathrm{test}\left(\mathrm{nm}\right)\times 100}{\mathrm{Absorbance}\;\mathrm{of}\;\mathrm{control}}$$

#### 2.10.4. Antibacterial Activity

A comparative antibacterial assay was performed against pathogenic bacterial strains using the disc-diffusion (Kirby–Baurer) method. All samples with three concentrations (25 µL/mL, 50 µL/mL, and 100 µL/mL) were prepared in distilled water. The experiment was carried out in triplicate. For sterilization, all the experimental instruments and nutrient agar were autoclaved at 121 °C and 17 lb pressure for one hour. The media was transferred into sterilized Petri dishes after sterilization and kept overnight at room temperature for solidification of media and to check for any contamination. A 10 µL bacterial suspension was poured on solidified media with the help of a micropipette and was spread with a glass spreader. Four paper discs were placed on the medium in each plate and subjected to different concentrations of leaf-AgNPs, bulb-AgNPs, and plant (leaf and bulb) extracts. The standard drug (streptomycin) and distilled water were used as +ve and −ve controls, respectively. All plates were incubated at 25 °C for 24 h. Antibacterial activity was observed by measuring inhibition zones with an ordinary scale [15,21]. 

#### 2.10.5. Antifungal Activity

Biofabricated silver nanoparticles and plant extract were comparatively assessed for antifungal properties against *Aspergillus niger*. Analysis was conducted following the well diffusion method [20]. All samples of AgNPs and plant extracts with three concentrations each (25 µL/mL, 50 µL/mL, 100 µL/mL) were prepared in distilled water. PDA (*Solanum tuberosum* L. dextrose agar) was prepared by adding nutrient agar and dextrose in Solanum tuberosum L. juice. All experimental equipment and PDA media were sterilized in an autoclave at 121 °C at 17 lb for one hour. To each sterilized plate, 20 mL molten media were transferred and left overnight to check for any contamination. In total, 10 fungal strains (*A. niger*) suspension were poured onto solidifying media using a micropipette and spread with a glass spreader. Four wells (5 mm each) were formed using a sterilized borer in each Petri dish and were labeled as a positive control, negative control, nanoparticles, and plant extracts, respectively. Various concentrations of all samples were poured into the respective wells. For positive and negative controls, clotrimazole (a standard antifungal drug) and distilled water were used, respectively. All Petri dishes were incubated at 25 °C for three days. Finally, the inhibition zones of the samples were measured using a scale and all zones were recorded in millimeters (mm). 

### 2.11. Analgesic Activity

To evaluate the significance of the tested samples and determine their capabilities to relieve pain, an acetic acid-induced writhing test following ARRIVE was conducted. The design of the study was approved by the Ethical Committee of Abdul Wali Khan University Mardan, KPK, Pakistan (AWKUM⁄Bot⁄2019⁄1672 (16 December 2019)). Swiss albino mice were evenly arranged into 6 groups. The experimental animals of the first group were considered the control group and injected with 1 mL of normal saline (1% *v*/*v*). The second group of mice was taken as the disease group and only injected with acetic acid. Then, 1 mL of acetic acid was injected into all mice of the third, fourth, fifth, and sixth groups. Ten minutes after then injection acetic acid solution, writhing was counted for 5 min. After writhing, the third group of mice were injected with 1 mL of paracetamol solution. The mice of the fourth, fifth, and sixth groups were subjected to 1 mL leaf ethanolic extract solution at doses of 50 mg/kg, 100 mg/kg, and 500 mg/kg, respectively. The same protocol was followed for bulb extract and leaf and bulb synthesized silver nanoparticles. After treatment with the samples, the inhibition of writhing was determined [20]. Finally, the percentage (%) inhibition rate of the tested samples was estimated using Equation (5).
(5)$$\%\;\mathrm{Inhibition}=\frac{\mathrm{No.}\;\mathrm{of}\;\mathrm{writhing}\;\mathrm{in}\;\mathrm{tested}\;\mathrm{samples}\times 100}{\mathrm{No.}\;\mathrm{of}\;\mathrm{writhing}\;\mathrm{in}\;\mathrm{control}}.$$

### 2.12. Establishment of In Vitro Micropropagation in Solanum tuberosum L.

Healthy internodal explants of the Astrix variety of *Solanum tuberosum* L. (potato) were collected from the “National Institute for Genomics and Advanced Biotechnology” (NIGAB), Islamabad (33°43′17.33″ N and longitude of 73°02′35.84″ E), Pakistan. *Solanum tuberosum* L. shoots with axillary buds were surface sterilized using 50% Clorox and 5.5% NaOCl, followed by washing with 70% ethanol and 1–2 drops of tween 20. The internodal explants were finally washed (3×) with distilled water for 10 min each. Uniformly sized axillary bud segments (2 cm each) enclosing the meristem were inoculated on Murashige and Skoog [22] media, 30 g−L sucrose, 6 g−L agars (Oxoid, Basingstoke, UK), pH 5.8 (pH 510, Eutech Instruments, Singapore) in sterilized glass vials. The cultures were incubated in a growth chamber (light intensity of 40–50 μmoL m^−2^ s^−1^ at 25 °C, photoperiod 16/8 h, relative humidity of 55–66%) until micro-propagation plantlets were grown. Uniform cuttings of each explant with two axillary buds were sub-cultured in an ethanol-sterilized chamber with laminar air flow and propagated in an MS medium supplemented with final concentrations of leaf-AuNPs and bulb-AgNPs (4 ppm each). Five cuttings of explants were sub-cultured in each glass jar, which were tightly sealed with lids and kept in the controlled conditions mentioned above. All sub-cultured plantlets were placed in a growth chamber until the day when phenotypic differences in biomass among the treatments were visibly noticed. Then, the plantlets were harvested after 30 days since first subculturing and used for subsequent experiments.

### 2.13. Evaluation for Biomass Production and Physiological Analysis

For the determination of fresh weight (FW), the aggregate of plantlets was harvested from all the respective treatments, washed with sterilized distilled water, and placed between filter papers to remove extra water. For dry weight (DW), each aggregate of plantlets was oven-dried (60 °C) for 24 h. 

### 2.14. Biochemical Evaluation of In Vitro Grown Plantlets of Solanum tuberosum L.

For total phenolic contents (TPC) determination, the Folin–Ciocalteu method was followed [23] using gallic acid as a positive control. The TFC was determined through the aluminum trichloride (AlCl_3_) method as described by Ref. [24]. Quercetin was used as a positive control. The results were expressed as μg Quercetin Equivalent (QE) per gram. Free radical scavenging activity was calculated as a percentage of DPPH° discoloration using the following Equation (6).
% Scavenging DPPH free radical = 100 × (1 − AE/AD) % scavenging DPPH free radical = 100 × (1 − AE/AD). (6)

In this equation, AE is the absorbance of the solution when a particular concentration of extract is added, and AD is the absorbance of the DPPH solution when nothing was added to this solution.

### 2.15. Analytical Method for Antioxidative Enzyme Activities

For antioxidative enzyme activities, the extract was prepared from fresh plant tissue by homogenizing with ice-cold 0.5 M Tris–HCl (pH 6.8) buffer, according to the method of Abbasi et al. [25]. Centrifugation was conducted at 10,000 rpm (20 min at 4 °C). Supernatant was used for enzyme assays. A UV–vis spectrophotometer (Halo DR-20, UV-vis spectrophotometer, Dynamica Ltd., Melbourne, VC, Australia) was used for measuring absorption. Phenylalanine ammonia lyase (PAL) activity (EC 4.3.1.5) was quantified as described by Ref. [26]. Catalase (CAT; EC 1.11.1.6) was quantified by the method of Ref. [27], superoxide dismutase (SOD; EC 1.15.1.1) was assayed by the method of Ref. [28], peroxidase (POD; EC 1.11.1.7) was determined as mentioned by Ref. [29], and ascorbate peroxidase (APx; EC 1.11.1.11) was determined by the method of Ref. [30].

### 2.16. Experimental Design and Statistical Analysis

Experiments were performed in a completely randomized design. One-way analysis of variance (ANOVA) using SPSS ver. 16.0 software (Chicago, IL, USA) was performed for data analysis (*p* ≤ 0.05).

## 3. Results and Discussion

### 3.1. Synthesis of Silver Nanoparticles

After adding leaf ethanolic extract to 1 mM silver nitrate solution, the color of the solution changed from light green to dark brown, indicating the synthesis of silver nanoparticles Figure 2A,B. Bulb aqueous extract also changed from whitish to dark brown when added to silver nitrate solution, thus indicating the fabrication of silver nanoparticles. Silver nanoparticles were previously synthesized by Ref. [31] (*Origanum vulgare*) and Ref. [32] (Pechual-loeschea leubnitzia). Production of AgNPs was carried out by Ref. [33] (*Allium sativum*) and Ref. [34] (*Allium cepa*) using bulb extract as a reducing and capping agent for silver ions. Changes in color of the sample solutions from light green and whitish to dark brown revealed AgNPs formation, as shown in Figure 3A–C. The formation of the brown color was due to the conversion of Ag+ ions into elemental silver with size ranges in nanometers. AgNPs synthesize by the bio reduction of Ag+ on exposure to metabolites, i.e., flavonoids, proteins, carbohydrates, enzymes, phenols, etc., present in plants which mainly reduce the metals to the nanoscale. The appearance of brown coloration is due to the excitation of surface plasmon resonance indicating silver nanoparticles fabrication, and it is the main characteristic of AgNPs [35]. 

### 3.2. Characterization

#### 3.2.1. UV-vis Spectroscopy and Optimization

The absorbency of reaction mixture with 1 mM aqueous Ag solution and ethanolic leaf extract of leaf and aqueous bulb extract of *A. vittata*, respectively, showed the fabrication of silver nanoparticles. The UV absorbance peak for different ratios for both leaf and bulb-capped AgNPs was obtained. For both leaf and bulb-capped AgNPs, maximum absorbance was observed in the ratio of 1:5, as shown in Figure 4A,B [36]. Temperature reaction has a significant role in controlling the configuration of nanoparticle nucleation processes. Different temperature conditions were applied to both leaf and bulb-capped AgNPs. It was found from the UV absorbance peak that both leaf and bulb-capped nanoparticles showed a maximum peak at 90 °C, as shown in Figure 4C,D [37]. The optimization of temperature on AgNPs formation was previously reported by Ref. [38] (*Allium cepa* and *Mussa acuminate*) and Ref. [39] (*Hippophea rhamnoides* Linn.). Their results observed that the favorable temperature for the biofabrication of silver nanoparticles was recorded as 40 °C and 75 °C, respectively. It was observed from leaf-capped AgNPs that with an increase in temperature in the range of 30–90 °C, the nanoparticle formation rate was increased, i.e., a maximum absorbency peak was observed at high temperatures with reduced size, as shown in Figure 4C. For bulb-capped AgNPs with an increase in temperature from 30–90 °C, the nanoparticle formation rate was increased, as shown in Figure 4D. The reason for particle size reduction and the high formation rate of nanoparticles with the increase in temperature is attributed to the rate of reaction, which enhances (speedup) with increased temperature, causing the consumption of Ag ions with faster speed, preventing the secondary reduction process on nuclei, and thus reducing the size of particles. Along with size reduction, the concentration of nanoparticles also increases with an increase in temperature [40]. Thermostable compounds in high temperatures play a significant role in the production of high yields of nanomaterials. The formation of small-sized nanoparticles at high temperatures is due to the prevention of aggregation of growing nanomaterial [39]. pH has an important role in the fabrication of AgNPs for controlling size and shape. The UV absorption spectra of AgNPs prepared from leaf extract showed that the absorption maximum was observed at pH 6, while for bulb-capped AgNPs, the absorption maximum was observed at pH 8. This shows that alkaline pH is suitable for nanoparticle synthesis for both leaf and bulb-capped AgNPs (Figure 4E,F), respectively [41]. The optimization of pH parameters on AgNPs using *Asalmalia malabaric* was conducted by Murali et al. [42], and their result indicated that pH 9 was favorable for the synthesis of silver nanoparticles. Synthesis of silver nanoparticles using green chemistry is affected by pH optimization due to the alteration of charges present on biomolecules of plant extracts. As Ag+ is a cation, the formation of AgNPs is also affected by the charges present on biomolecules [43]. Due to the presence of a high concentration of OH (hydroxyl) groups on the surface of nanoparticles in alkaline pH, the size of nanoparticles reduces because of dominant repulsive forces in colloidal solutions, which lead to a reduction of particle aggregation. At acidic pH, due to the aggregation of AgNPs, large-sized nanoparticle formation occurs. On the other hand, in the case of alkaline pH, there are many functional groups available to bind and cap silver ions preferring the production of several small-sized silver nanomaterials (Figure 4E,F), as shown by Ahmad et al. [39]. NaCl concentration also has significant effects on silver nanoparticle synthesis of controlled size and shape. Salt solutions of various concentrations such as 0.2 and 0.4 mL were added to the prepared AgNPs of leaf and bulb extract of *A. vittata*, respectively. For both leaf and bulb-capped AgNPs, it was found from UV absorbance that with a higher concentration of NaCl, AgNPs showed low stability, and their stability was affected (Figure 4G,H) [44]. *Salmalia malabaric* plant extract was previously demonstrated by Ref. [42] for the optimization of salt concentration effects on the biogenesis of silver nanoparticles. Their result found that 5 M of NaCl (sodium chloride) was favorable for nanoparticle formation because maximum peak absorbency was observed at this particular molar concentration. The effect of sodium chloride on AgNP formation was also reported by Ref. [45], and their study revealed that 30 mM NaCl was recorded as the optimum salt concentration for silver nanoparticle synthesis. Our result showed that as the concentration of 1 mM sodium chloride solution was increased, the formation rate of silver nanoparticles was gradually decreased, i.e., the maximum peak wavelength was recorded at 0.2 mL but as the concentration increased from 0.2 to 1 mL the absorbency peak was reduced such that minimum absorbance was observed at 1 mL NaCl solution, as shown in Figure 4G,H. Additionally, maximum absorbance for AgNPs was observed for the optimized ratio, i.e., without the addition of NaCl. This result is due to the presence of Cl^−1^ ions: as the concentration of NaCl increases, the accumulation of Cl ions also increases in reaction, due to which the formation of large-sized particles occurs because Cl^−1^ ions encourage nuclei particles to grow rapidly in reaction. It was also observed that the reaction rate of leaf-capped AgNPs was decreased as NaCl concentration was gradually increased, i.e., the change in a shift in leaf-AgNPs was increased from 450 nm to 550 nm, as shown in Figure 4G. The reason for the change in a shift in absorbency wavelength is because Cl^−1^ ions accumulate around nuclei particles in solution and increase the size of particles thus reducing the reaction rate of AgNPs, Gautam et al. [45]

#### 3.2.2. FT-IR and X-rays Diffraction (XRD) Analysis

The biogenic AgNPs were synthesized from biological moieties for the FTIR measurements to identify the ethanolic leaf extract and aqueous bulb extract biomolecules and their possible involvement. The FTIR spectra of dried leaf-capped AgNPs are shown in Figure 5A. The FTIR spectra of leaf-capped AgNPs showed peaks at 3287, 2921, 1622, and 1016 cm^−1^ (Figure 5A). The results showed that shifts at 3287 cm^−1^ were allocated to the N-H group, 2921 cm^−1^ for the C-H bond, and 1622.3 cm^−1^ and 1016 cm^−1^ shifts were due to C=C and C-O, respectively [46] (Figure 5A). The carbonyl groups proved the presence of flavanones that are adsorbed on the surface of metal nano-sized particles by interaction through π-electrons in the carbonyl groups in the absence of a sufficient concentration of chelating agents. It was also confirmed that the carbonyl group from the protein and amino acid had a stronger ability to bind with metal nanoparticles or act as capping and stabilizing agents. Similarly, in the frame of the current study, the FTIR spectra of dried bulb-capped AgNPs show peaks at 3280, 2921,1638, and 1016 cm^−1^ (Figure 5B). It was concluded that bands at 3280 cm^−1^ were allocated to the N-H group, the band at 2921 cm^−1^ showed a C-H bond, 1638 cm^−1^ was allocated to C=O, and the shift at 1016 cm^−1^ was due to the C-O group, as described by Massey et al. [46]. An aqueous solution of *A. vittata* was phytochemically tested to determine its active compounds. The phytochemical analysis of *A. vittata* showed the presence of flavonoids, alkaloids, and coumarins [47]. The intensities of the spectral peaks of both leaf-capped AgNPs as well as bulb-capped AgNPs were reduced compared to those of plant extract as well as bulb extract, respectively. The results indicate that at spectral bands of AgNPs where peak intensity was reduced or disappeared, the functional groups (phytochemicals) present in that region were generated by the subsequent oxidation involved in reducing silver ions and forming nanoparticles [48]. Additionally, the presence of carbonyl groups proved the presence of flavanones that are adsorbed on the surface of metal nano-sized particles by interaction through π-electrons in the carbonyl groups in the absence of a sufficient concentration of chelating agents. It was also confirmed that the carbonyl group from the protein and amino acid had a stronger ability to bind with metal nanoparticles or act as capping and stabilizing agents [49] (Figure 5A,B). X-ray diffraction crystallography is a technique used to identify the molecular and atomic structure of crystals, in which crystal-shaped atoms cause a beam of accident X-rays to diffract in many specific directions. The crystallinity of dried AgNPs capped by leaf and bulb extracts of *A. vittata* was confirmed, respectively, with the help of X-ray diffraction. The X-ray diffraction pattern of silver nanoparticles showed the Braggs representation of the face center cubic structure of silver. The characteristic diffraction peaks of fcc silver lattice for bulb synthesized nanoparticle were at 37.9, 46.13, 64.25, and 77.16 in 2Ø with corresponding planes at (111), (200), (220), and (311), respectively. For leaf extract mediated nanoparticles, the XRD diffraction pattern showed diffraction peaks at 37.9, 46.1, 64.61, and 76.9 in 2Ø with corresponding planes at 111, 200, 220, and 311, respectively [50] (Figure 5D). The crystalline nature of bulb and leaf fabricated silver nanoparticles were confirmed by XRD analysis. The formation of lattice planes of nanoparticles was indicated by the XRD pattern. The presence of small-sized nanomaterial and no evidence of the presence of bulk residue (remnant) and impurity were expressed by the peak broadening of this pattern [21] (Figure 5C,D).

#### 3.2.3. Scanning Electron Microscopy (SEM) and Energy Dispersion Spectroscopy

Silver nanoparticles synthesized using leaf and bulb extracts of *A. vittata* were subjected to SEM to estimate and analyze shape and surface morphology at different magnifications. It was observed from the nanoparticle micrograph that both bulb-AgNPs and leaf-AgNPs are spherical, monodispersed, and well distributed. It is also indicated from the micrograph result that particle aggregation is also observed. Leaf-AgNPs and bulb-AgNPs were analyzed for elementary composition using energy dispersion spectroscopy. EDS analysis confirmed the presence of AgNPs at 3Kve for both leaf and bulb-capped nanoparticles. It was also indicated that other elements, i.e., C, Si, N, and Cl, are also present. Leaf- and bulb-capped nanoparticles also revealed that a high peak was observed for Ag (Figure 6).

SEM analysis of biosynthesized silver nanoparticles was also reported by Devi et al. [51] using *Euphorbia hirta* leaf extract, and their work concluded the presence of spherical and cubic-shaped silver nanoparticles. Analysis of scanning electron microscopy reported the presence of spherical-shaped and monodispersed leaf-AgNPs and bulb-AgNPs. The reduction process held at the nanoparticle surface was also determined by micrograph of SEM analysis, as shown in Figure 6A,B. According to [39], the presence of agglomeration in some particles is because of Van der Waals forces (intermolecular interactions) between these nanoparticles. Energy dispersion spectroscopy of silver nanoparticles was previously reported using Pechual-loeschea leubnitzia, as stated by Mofolo et al. [32], and *Vitis vinifera* [52]. Their results investigated the presence of elemental Ag as well as carbon, nitrogen, and oxygen. Their results follow our results for EDX analysis, as shown in Figure 6C,D.

### 3.3. Biological Significance

#### 3.3.1. Insecticidal Potential

Insecticidal activity evaluation was carried out using silver nanoparticles prepared from leaf and bulb extract using *A. vittata* plants against *Tribolium castenium* at different concentrations, i.e., 100, 500, and 1000. It was observed that silver nanoparticles are more effective and potentially applicable against *Tribolium castenium* than extracts. The result indicated that the highest mortality rate was significantly (*p* < 0.05) observed in bulb-capped AgNPs at 1000 µg/mL with a 56.6% mortality rate, followed by leaf-capped nanoparticles which show the highest significant (*p* < 0.01) mortality rate of 50% at 1000 µg/mL. Leaf extract showed a significant (*p* < 0.05) mortality rate of 46.67% at 1000 µg/mL, while bulb aqueous extract showed a significant (*p* < 0.05) mortality rate of 43.34% at 1000 µg/mL (Figure 7A). Using *Tribolium castaneum* (wheat insect), analysis for the evaluation of the insecticidal activity of silver nanoparticles was carried out using malathion [16] and *Myriostachya wightiana* [53]. The results reported that these silver nanoparticles showed 100% and 55.2% mortality rates, respectively. Our result observed a 56.7% and 50% mortality rate for bulb and leaf extract fabricated silver nanoparticles, respectively, as shown in Figure 7A. AgNPs cause dehydration of insect integument and, with the help of abrasion and sorption, cause damage to the cuticle layer by destroying the protective wax present in the cuticle. It was also revealed that blockage of the trachea and spiracles of insects occurs due to the impairment of the alimentary canal on exposure to AgNPs, which leads to mortality, as documented by Rouhani et al. [54].

#### 3.3.2. Phytotoxic Assessment 

The present study determined the phytotoxic potential of bulb- and leaf-capped AgNPs of *A. vittata* at 50 µg/mL, 100 µg/mL, and 500 µg/mL concentrations. All samples showed a significant inhibition rate to Lemna aequincotialis. It was observed that the highest inhibition rate with significance (*p* < 0.0001) was observed by bulb-capped silver nanoparticles (96.6%) at 500 µg/mL, followed by Atrazine (standard herbicide) which non-significantly showed 93.3% inhibition at 500 µg/mL. Bulb aqueous extract as compared to bulb-AgNPs significantly (*p* < 0.001) showed an 86.6% inhibition rate at 500 µg/mL. It was also concluded from the result that leaf-capped AgNPs showed the highest significant (*p* < 0.0001) inhibition rate of 86.6% at 500 µL/mL while leaf extract showed a significant (*p* < 0.001) 70% inhibition rate at 500 µg/mL (Figure 7B). Phytotoxic activity analysis of silver nanoparticles prepared by plant extracts of *Malcolmia cabulica* and *Bistorta affinis* was carried out by Sultana et al. [17] and inhibition rates of 85% and 67%, respectively, were concluded. As can be seen in Pereira et al. [55], the phytotoxic activity of nanoparticles using *Lemna* minor as test plants was also reported. Their work attributed to the good phytotoxic potential of these particles. Bulb- and leaf-capped silver nanoparticles exhibited good inhibition properties against *Lemna aequincotialis*, as shown in Figure 7B. According to Dewez et al. [56] and Sultana et al. [17], the inhibition of *Lemna* plants is due to a disturbance of plant physiological properties. The nanoparticles produce stimulants that affect the synthesis of chlorophyll and deteriorate photosynthesis by changing the biomass of the *Lemna* plant. Therefore, it was concluded from the above-mentioned study that the production of these stimulants by silver nanoparticles would be formulated with herbicides for the improvement of crop yields, Sultana et al. [17].

#### 3.3.3. Antioxidant Potential 

The highest ability of DPPH radical inhibition was found to be significant (*p* < 0.0001) in leaf-synthesized silver nanoparticles with 92% inhibition at 100 µg/1 mL, followed by ascorbic acid, which showed the highest significant (*p* < 0.0001) inhibition of 87%. Nanoparticles synthesized using aqueous bulb extract showed the highest significant (*p* < 0.0001) inhibition rate of 69.6 at 100 µg/1 mL compared to aqueous bulb extract, which showed a significant (*p* < 0.0001) maximum inhibition rate of 48.7% (Figure 7C). The antioxidant activity of AgNPs synthesized using *Cesturnum nocturnum* was confirmed by Keshari et al. [21], and their investigation determined that silver nanoparticles compared with vitamin C showed good scavenging properties, as shown in Figure 7C. A free radicals scavenging assay was also performed on *Dryopteris blanfordii* by Khan et al. [57]. Silver nanoparticles capped with bulb and leaf extracts of *A. vittata* exhibited significant antioxidant properties. The antioxidant potential of AgNPs is revealed due to the presence of bioactive molecules (functional groups) on the surface of these nanoparticles causing the reduction and capping of silver ions. These functional groups play an active role in neutralizing free radicals documented by Keshari et al. [21]. According to Sultana et al. [17], this result is also attributed to the fact that biologically capped nanomaterials exhibit antioxidant agents, which decrease the absorbance by donating protons. It was determined that low absorbency is attributed to high antioxidant potential. 

#### 3.3.4. Antibacterial Capacity

The antibacterial potency of prepared silver nanoparticles using leaf and bulb extracts of the *A. vittata* plant was evaluated using the disc diffusion method against *Staphylococcus aureus* and *Pseudomonas aeruginosa.* The potential was calculated by the presence or absence of inhibition zones measured in millimeters. It was observed by the fact that biofabricated silver nanoparticles showed a significant zone of inhibition as compared to plant (leaf and bulb) extracts at 100 µL/mL (Figure 7D). This activity showed that leaf-capped AgNPs were observed as significant (*p* < 0.0001), showing the highest zone of inhibition (20 mm) against *Staphylococcus aureus* and lowest inhibition (16 mm) against *Pseudomonas aureginosa*. This is significant (*p* < 0.0001) compared to leaf extract, which showed a high inhibition zone (18 mm) against *Staphylococcus aureus* while showing no inhibition against *Pseudomonas aureginosa*. Silver nanoparticles prepared from bulb extract showed a significant (*p* < 0.0001) high inhibition zone (19 mm) against *Pseudomonas aureginosa* (Figure 8A) and 16 mm against *Staphylococcus aureus*. Only aqueous bulb extract showed a significant (*p* < 0.0001) 14 mm inhibition zone against *Staphylococcus aureus* while showing no inhibitory zone against *Pseudomonas aureginosa* (Figure 8B). Streptomycin, which was used as a standard antibiotic drug, showed a significant (*p* < 0.0001) maximum zone of inhibition (24 mm) against *Staphylococcus aureus* and 22 mm against *Pseudomonas aureginosa* compared to plant extracts and silver nanoparticles (Figure 8C). It was concluded from the above result that biofabricated silver nanoparticles have significant potential against bacteria. 

The antibacterial activity of silver nanoparticles synthesized by *Allium cepa* extracts was evaluated against *E. coli* and *Salmonella typhimurium* (pathogenic bacterial strains), and it was concluded that these nanoparticles have significant potential against these pathogenic strains as reflected in the literature of Rao et al. [58] who worked on the antibacterial activity of silver nanoparticles against *Klebsiella pneumonia*, *Pseudomonas aeruginosa*, and *Escherichia coli*. The study observed that the highest inhibition was showed by *Klebsiella pneumonia* (15 mm zone of inhibition) *E. coli* and *P. aeruginosa* showed 10 mm and 8 mm inhibition, respectively. Leaf-AgNPs and bulb-AgNPs showed good antibacterial potential against pathogenic bacterial strains, as shown in Figure 8D. This result is supported by the reason provided by Balamanikandan et al. [59], that silver nanoparticles enter bacterial cells and damage bacteria by condensing their DNA molecules, thus halting cell division because the replication of DNA takes place in a relaxed state and is destructed in condensed conditions. In addition to the condensation of DNA, silver ions inactivate bacterial cell protein by attaching to the thiol group, causing these proteins to denature. According to Park et al. [43], damage to bacterial cells by the penetration of cell walls increases cell permeability, leading to cell death. 

#### 3.3.5. *Antifungal Significance*

The antifungal potency of prepared silver nanoparticles using leaf and bulb extracts (Figure 9A) of the *A. vittata* plant was evaluated by a well diffusion method against *Aspergillus niger*. The potential was calculated by the presence or absence of inhibition zones measured in millimeters. It was observed that biofabricated silver nanoparticles showed a significant zone of inhibition compared to plant (leaf and bulb) extracts. This activity showed that bulb-synthesized silver nanoparticles significantly (*p* < 0.0001) showed the highest zone of inhibition (15 mm) at 100 µL/mL and 12 mm at 50 µL/mL against *Aspergillus niger*, followed by clotrimazole (a standard antifungal drug) which showed a 12 mm maximum zone of significant (*p* < 0.0001) inhibition. Bulb extract showed a 3 mm significant (*p* < 0.0001) maximum zone of inhibition at 100 µL/mL. Silver nanoparticles prepared using leaf extract showed a significantly (*p* < 0.0001) high inhibition zone (11 mm) at 100 µL/mL and 9 mm at 50 µL/mL as in Figure 9B. This was significant (*p* < 0.0001) against *Aspergillus niger* compared to leaf extract which showed a significant (*p* < 0.0001) maximum zone of inhibition of 5 mm at 100 µL/mL (Figure 9B). Jamdagni et al. [40] carried out an antifungal assay of silver nanoparticles prepared using *Elettaria cardamomum* leaf extract against *Aspergillus niger*. Their studies revealed that these silver nanoparticles were effective against this pathogenic fungal strain. Biofabricated silver nanoparticles using *Allium cepa* extract as a reducing agent were evaluated by Ref. [59] for antifungal properties against *Aspergillus* species, i.e., *Aspergillus niger*, *Aspergillus terreus*, *Aspergillus flavus*, and *Aspergillus ochraceous*. The result reported that all fungal strains were effectively inhibited by these nanoparticles. Our antifungal result concluded that these bulbs and leaf-capped AgNPs exhibit good antifungal properties Figure 7E. This is due to the large surface area to volume ratio of these silver nanoparticles because as the surface area to volume increases, the interaction of nanoparticles to fungal cells increases, which causes damage to the structure of fungal cells by inactivating protein and condensing DNA molecules, leading to cell death [43,59]. It was determined that silver nanoparticles, in general, disturb cell wall permeability by attaching to the cell wall and affecting the respiration process of the cell. It was also revealed that these nanoparticles destroy cell structure by interacting with phosphorous and sulfur-containing compounds, i.e., protein and DNA [60].

#### 3.3.6. Analgesic Capabilities 

The highest significant (*p* < 0.05) percentage of inhibition was shown by leaf-synthesized silver nanoparticles (93%) at 500 µL/mL in which writhing was reduced to 2.75 ± 0.75 * followed by standard drug paracetamol, which showed significant (*p* < 0.05) inhibition of 76.77%, in which writhing was reduced to 9 ± 3.0 **. Leaf ethanolic extract showed the highest significant (*p* < 0.05) inhibition rate of 80.64% at 500 µg/mL, which reduced the writhing rate to 7.5 ± 1.19 **. In a comparative study of bulb extract and bulb-capped nanoparticles, the highest %inhibition with significance (*p* < 0.05) was observed by bulb nanoparticles (91%), while bulb extract showed the highest significant (*p* < 0.05) inhibition of 77.7%, reducing writhing to 3.5 ± 2.5 * and 8.75 ± 1.88 **, respectively (Figure 7F). Analgesic activity analysis of AgNPs using the Buchu plant was performed by Chiguvare et al. [61], and their result showed that these fabricated silver nanoparticles possess effective pain revealing properties compared to aspirin, as stated by Shah et al. [62], who studied the analgesic activity of *Dryopteris blanfordii.* Their result showed 45% inhibition compared to the standard drug (Aspirin), which showed 40.43% inhibition. As compared to the plant extract of *A. vittata*, silver nanoparticles biofabricated using these extracts determined effective analgesic potential. Additionally, this study revealed the active role of synergistic potential of plant secondary metabolites and AgNPs.

### 3.4. In Vitro Growth and Physiological Response of Solanum tuberosum L. upon AgNPs Supplementation

In vitro micropropagation, growth response, and biomass production of Solanum tuberosum L. supplemented with leaf-AgNPs (4 ppm) and bulb-AgNPs (4 ppm) were evaluated in MS media. Both types of biogenic AgNPs positively induced micropropagation and biomass production in terms of an overall increase in shoot length and fresh and dry weight compared to the control (without AgNPs) (Figure 10A–C). Shoot length was significantly (*p* < 0.05) increased upon supplementing the leaf-AgNPs (8.1 ± 0.4 cm) and bulb-AgNPs (9.4 ± 0.7 cm) compared to the control (5.9 ± 0.5 cm). Fresh weight was significantly (*p* < 0.05) increased upon supplementing the leaf-AgNPs (2.85 ± 0.09 g) and bulb-AgNPs (3.54 ± 0.07 g) compared to the control (1.6 ± 1.25 g). Dry weight was significantly (*p* < 0.05) increased upon supplementing the leaf-AgNPs (0.65 ± 0.004 g) and bulb-AgNPs (0.082 ± 0.005 g) compared to the control (04 ± 0.003 g). Similarly, the MS media supplemented with leaf-AgNPs and bulb-AgNPs showed significantly (*p* < 0.05) higher total flavonoid content (TFC), total phenolic content (TPC), and DPPH free radical scavenging activity compared to the control (Figure 10D–F). TFC was significantly (*p* < 0.05) increased upon supplementing the leaf-AgNPs (3.05 ± 0.155 µg/mL) and bulb-AgNPs (3.5 ± 0.2 µg/mL) compared to the control (2.08 ± 0.27 µg/mL). TPC was significantly (*p* < 0.05) increased upon supplementing the leaf-AgNPs (1.4 ± 0.044 µg/mL) and bulb-AgNPs (2 ± 0.09 µg/mL) compared to the control (0.703 ± 0.04 µg/mL). DPPH free radical scavenging activity (%) was significantly (*p* < 0.05) increased upon supplementing the leaf-AgNPs (80 ± 3.8 µg/mL) and bulb-AgNPs (88 ± 3.9 µg/mL) compared to the control (52 ± 3.3 µg/mL).

Antioxidative enzymes such as phenylalanine ammonia-lyase (PAL), superoxide dismutase (SOD), catalase (CAT), ascorbate peroxidase (APX), and peroxidase (POD) were significantly (*p* < 0.05) increased in in vitro growth plantlets of *Solanum tuberosum* L. supplemented with leaf-AgNPs and bulb-AgNPs compared to control cultures (Figure 10). Phenylalanine ammonia-lyase (PAL) activity increased upon supplementing the leaf-AgNPs and bulb-AgNPs with up to 15 ± 0.28 U/mg protein and 17 ± 0.4 U/mg protein compared to the control culture showing PAL activity up to 5.8 ± 0.3 U/mg protein. Catalase (CAT) activity was increased upon supplementing the leaf-AgNPs with up to 3.6 ± 2.5 U/mg protein and bulb-AgNPs with up to 2.5 ± 0.2 U/mg protein compared to the control (0.7 ± 0.2 U/mg protein). Superoxide dismutase (SOD) was significantly (*p* < 0.05) increased upon supplementing the leaf-AgNPs (6.9 ± 3.8 U/mg protein) and bulb-AgNPs (7.2 ± 0.34 U/mg protein) compared to the control (3 ± 0.49 U/mg protein). Peroxidase (POD) was significantly (*p* < 0.05) increased upon supplementing the leaf-AgNPs (4.3 ± 0.22 µg/mL) and bulb-AgNPs (6.5 ± 0.24 U/mg protein) compared to the control (1.8 ± 0.35 U/mg protein). Ascorbate peroxidase (APX) was significantly (*p* < 0.05) increased upon supplementing the leaf-AgNPs (4.8 ± 0.23 U/mg protein) and bulb-AgNPs (4.6 ± 1.3 U/mg protein) compared to the control (2.7 ± 0.48 U/mg protein) in Figure 11. Among the different types of nanomaterials, silver nanoparticles (AgNPs) have received global attention due to their tremendous physiological properties, including higher antimicrobial potential. Furthermore, AgNPs have been used widely for their pertinent potential in influencing plant cell growth, biomass production, and induction of bioactive secondary metabolites in plant cell cultures. Using AgNPs either solely in growth media or augmented with plant growth regulators (PGRs), such as α-naphthalene acetic acid (NAA), profoundly influenced the callus growth and antioxidant potential in plant tissue cultures [31]. Nanoparticles are among the latest currently studied elicitors as they have a considerable impact on the physiological processes of plants, such as seed germination, growth, biomass, and metabolism [63,64]. *Solanum tuberosum* L.es produce bioactive and defense-related secondary metabolites, including glycoalkaloids, calystegines alkaloids, Steroidal Alkaloids, protease inhibitors, lectins, phenolic compounds, flavonoids, and antioxidant enzymes. These bioactive secondary metabolites have either beneficial effects on diet or show phytopathogenic effects on plant survival upon infections as stated by Kaunda et al. [65].

Our findings indicate that these bulb- and leaf-synthesized biogenic AgNPs are efficient elicitors for biomass enhancement and beneficial secondary metabolic induction in *Solanum tuberosum* L. tissue culture at optimum concentration. 

## 4. Conclusions

We used the green and simple procedure for the synthesis of AgNPs at atmospheric pressure and room temperature. The whole method is said to be “green” because of the reduced energy consumption required (reaction at atmospheric pressure and room temperature). *Amaryllis vittata* (L.) Herit is an effective source for the fabrication of AgNPs. This was confirmed by the conversion of the green color of leaf ethanolic extract and the whitish color of aqueous bulb extract to brown color. The leaf and bulb extract of *A. vittata* plants has great capability for reducing silver nitrate solution, resulting in the formation of silver nanoparticles. The UV-vis spectroscopy of the solution with leaf extract and silver nitrate solution, as well as bulb aqueous extract and silver nitrate solution, observed maximum absorbency in the range of 400–500 nm, which confirms the formation of AgNPs. X-ray diffraction analysis indicated that nanoparticles are face-centered cubic (FCC) in structure. The generated nanoparticles were found to be crystalline spherical and biofunctionalized with organic molecules. Energy dispersion spectroscopy indicated the confirmation of the presence of the Ag element. The highest insecticidal and phytotoxic activities were recorded in leaf-AgNPs and bulb-AgNPs nanoparticles using *A. vittata* along with antimicrobial potential against pathogenic microbes. Both the tested samples offered significant potential to scavenge the free radicals. The acetic acid-induced writhing activity confirmed that leaf-AgNPs and bulb-AgNPs have the potential for analgesic properties. It was concluded from the above research that these biofabricated nanoparticles have significant pharmacological importance and can be used in designing and processing new drugs used in the pharmacological and agricultural fields. Additionally, this study proved an emerging application of nanotechnology in agriculture for the establishment of viable biomass production and beneficial secondary metabolite induction.

## Figures and Tables

**Figure 1 materials-15-05478-f001:**
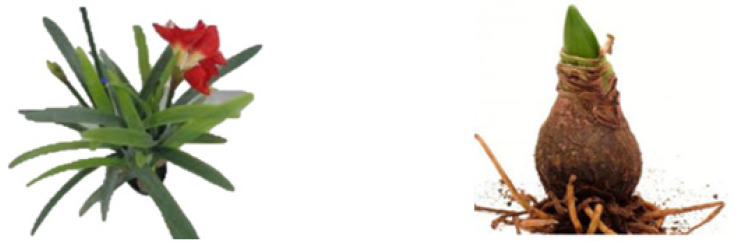
Leaf and Bulb of *Amaryllis vittata* (L.) Herit.

**Figure 2 materials-15-05478-f002:**
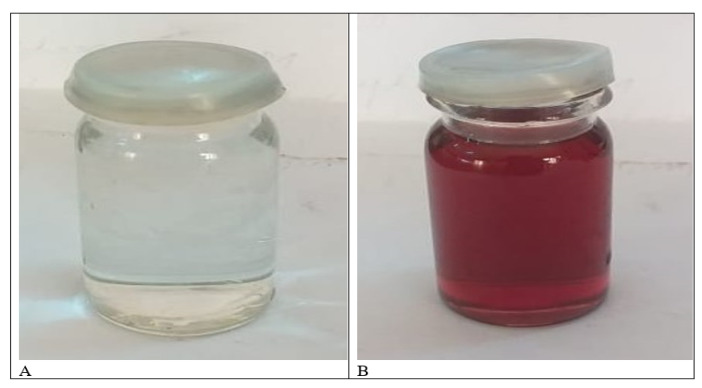
(**A**) Bulb extract + AgNO_3_ solution before reaction. (**B**) Bulb extract + AgNO_3_ solution after reaction.

**Figure 3 materials-15-05478-f003:**
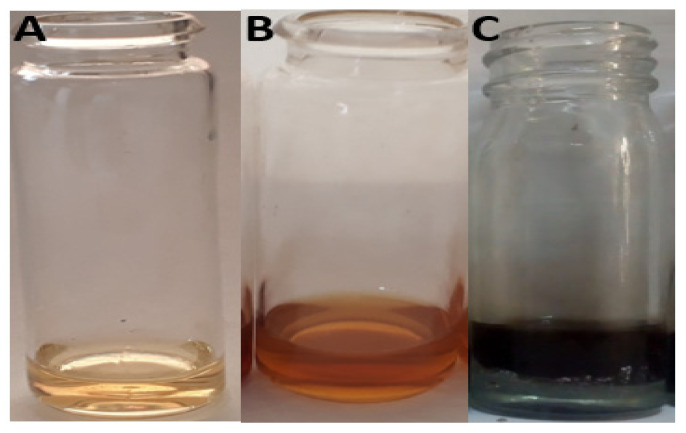
(**A**) Leaf extract + AgNO_3_ solution before reaction, (**B**) Leaf extract + AgNO_3_ solution after 1 h, and (**C**) Leaf extract + AgNO_3_ after 24 h.

**Figure 4 materials-15-05478-f004:**
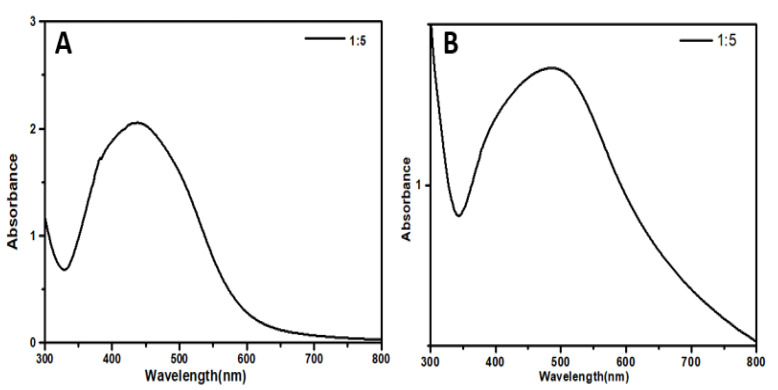
(**A**,**B**) UV-visible spectra of AgNPs/*A. vittata* having different AgNO_3_/Plant ratios, (**C**,**D**) effect of pH on the stability of AgNPs, (**E**,**F**) effect of temperature on the stability of AgNPs, and (**G**,**H**) effect of salt (NaCl) on the stability of AgNPs.

**Figure 5 materials-15-05478-f005:**
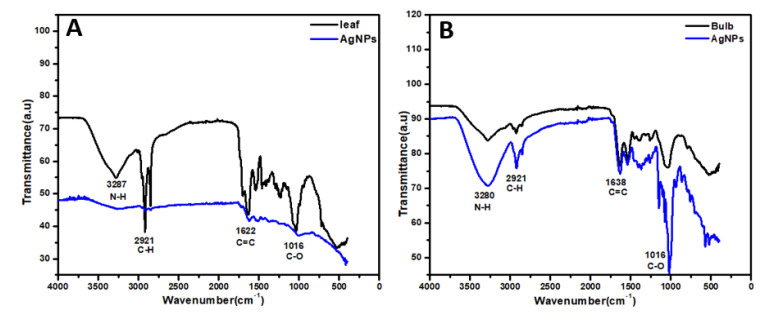
FTIR of leaf-capped AgNP (**A**) and bulb-capped AgNP (**B**) extracts of *A. vittata*. XRD analysis of bulb (**C**) and leaf synthesized AgNPs (**D**) using *A. vittata*.

**Figure 6 materials-15-05478-f006:**
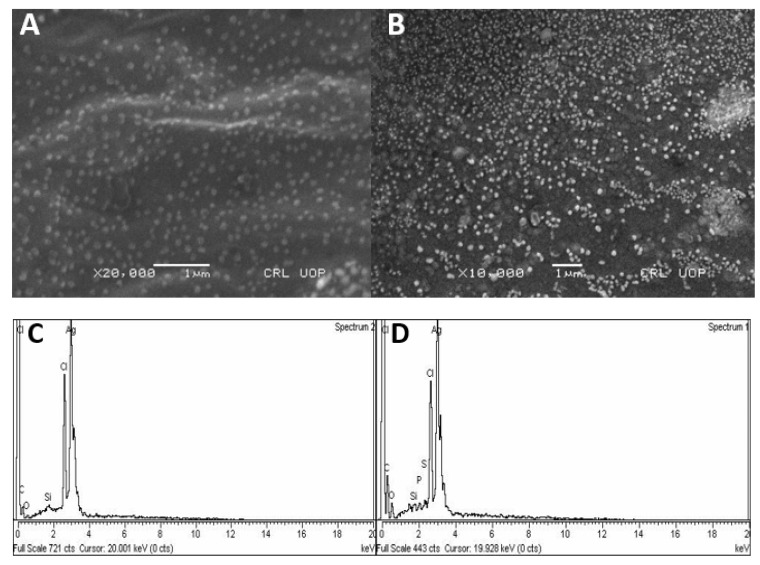
Micrographs of SEM analysis of leaf-capped AgNPs (**A**) and bulb-capped AgNPs (**B**) at 1 µm magnification. EDX of leaf-capped AgNPs (**C**) and bulb-capped AgNPs (**D**).

**Figure 7 materials-15-05478-f007:**
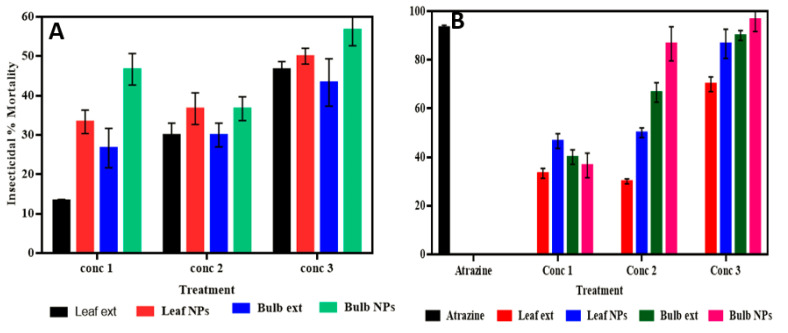
Insecticidal (**A**), phytotoxic (**B**), antioxidant (**C**), antibacterial (**D**), antifungal (**E**), and analgesic (**F**) activity of leaf-capped AgNPs and bulb-capped AgNPs.

**Figure 8 materials-15-05478-f008:**
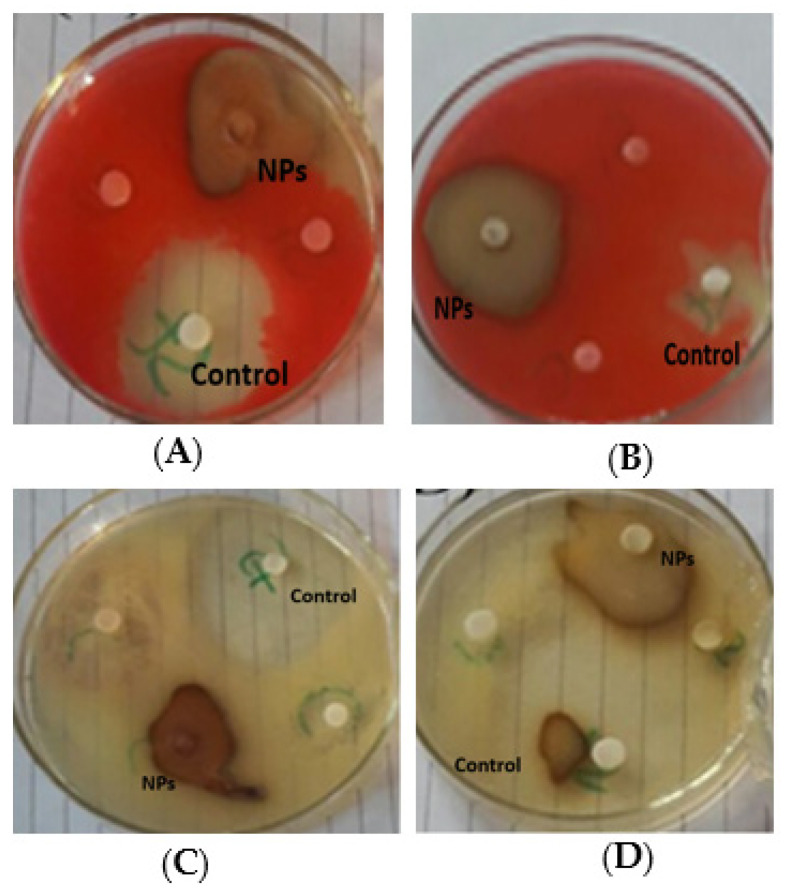
Antibacterial activity of leaf- and bulb-capped AgNPs and plant (leaf and bulb) extract against *Staphylococcus aureus* (**A**,**B**) and *Pseudomonas aureginosa* (**C**,**D**).

**Figure 9 materials-15-05478-f009:**
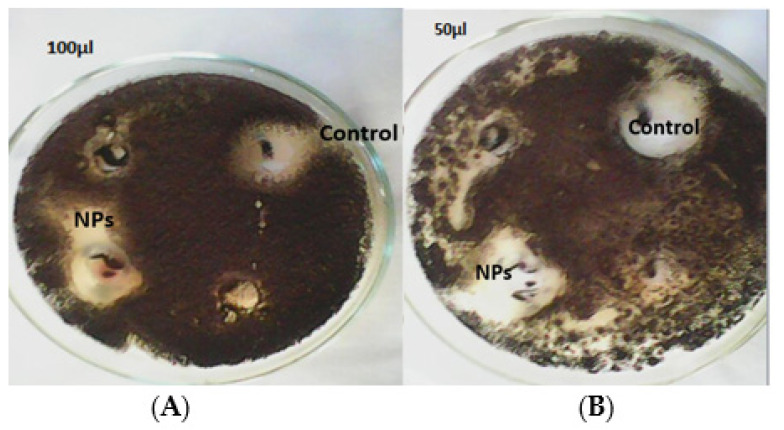
Antifungal activity of (**A**) bulb-capped AgNPs and (**B**) leaf-capped AgNPs against *Aspergillus niger*.

**Figure 10 materials-15-05478-f010:**
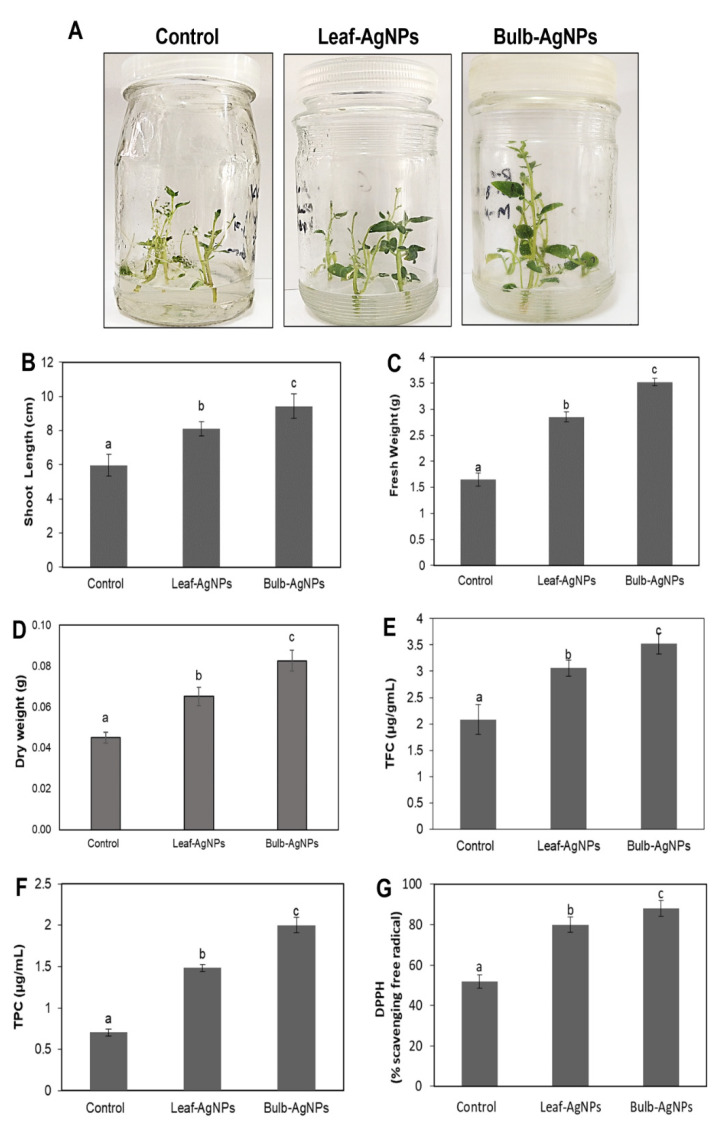
Effects of leaf-AgNPs and bulb-AgNPs regarding the in vitro micropropagation, growth response, and biomass production of Solanum tuberosum L. plantlets grown under a controlled environment. (**A**) Comparison of plant height, (**B**) shoot length, (**C**) fresh weight, (**D**) dry weight, (**E**) total flavonoid content, (**F**) total phenolic content, and (**G**) DPPH (% free radical scavenging activity) at 30 DAC. Values are the mean ± standard error from three replicates. Different letters indicate a significant difference (*p* < 0.05) in values between control and treated plantlets.

**Figure 11 materials-15-05478-f011:**
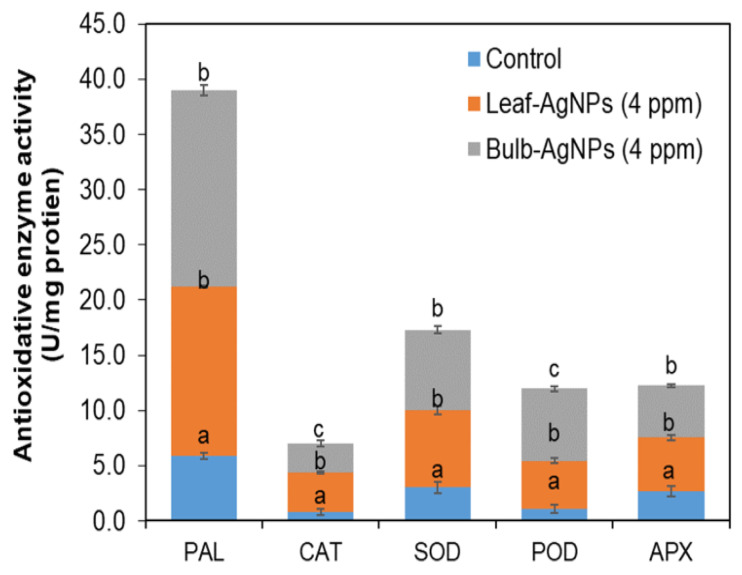
Effects of leaf-AgNPs and bulb-AgNPs on activities of antioxidative enzymes (PAL, CAT, SOD, POD, and APX, in U/mg protein) in *Solanum tuberosum L.* plantlets grown under a controlled environment in in vitro conditions. Values are the mean ± standard error from three replicates. Different letters indicate a significant difference (*p* < 0.05) in values between control and treated plantlets.

## Data Availability

Not applicable.

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
