# Peer review of "Biological Synthesis of Silver Nanoparticles by Amaryllis vittata (L.) Herit: From Antimicrobial to Biomedical Applications"

_materials, 2022, doi:10.3390/ma15165478_

Round 1

Reviewer 1 Report

Dear author, please revise the manuscript as following suggested points.

1. Draw a schematic figure that should express the intention of the whole story of your paper at a glance.

2. Cite the latest silver nanoparticles-based suggested article in the introduction. Silver-loaded carboxymethyl cellulose nonwoven sheet with controlled counterions for infected wound healing." Carbohydrate Polymers 286 (2022): 119289.

3. Replace figures 5 and 6 with better quality pictures these are not suitable for a publication.

4. revise figure 8A the text should be on the side of the box.

5. retake the picture for figure 3 in a small caped tube or vials that should look better than the current one.

6.in Figure 3. FTIR of leaf capped AgNPs (A) and bulb capped AgNPs (B) extract of A. vittata, XRD 479 analysis of Bulb (C) and Leaf synthesized AgNPs (D) using A. vittata. ( give full length of FTIR spectra with zoom image.

7. For antibacterial assay cite the following article Composite Anion Exchange Membranes with Antibacterial Properties for Desalination and Fluoride Ion Removal." ACS ES&T Water 1, no. 10 (2021): 2206-2216.

Author Response

Reviewer 1

Comments and Suggestions for Authors

Dear author, please revise the manuscript as following suggested points.

Author response: Worthy reviewer thanks for considering our work and comments and suggestion for the improvement of our article.

Reviewer Comment: Draw a schematic figure that should express the intention of the whole story of your paper at a glance.

Author response: Dear reviewer as per your suggestion the abstract which reflects our project has been provided as graphical abstract in the revised version.

Reviewer Comment: Cite the latest silver nanoparticles-based suggested article in the introduction. Silver-loaded carboxymethyl cellulose nonwoven sheet with controlled counterions for infected wound healing." Carbohydrate Polymers 286 (2022): 119289.

Author response: Very informative and added in the revised version.

Reviewer Comment: Replace figures 5 and 6 with better quality pictures, these are not suitable for publication.

Author response: The figure is changed as suggested by the reviewer.

Reviewer Comment: revise figure 8A the text should be on the side of the box.

Author response: The figure is changed as suggested by the reviewer.

  1. retake the picture for figure 3 in a small caped tube or vials that should look better than the current one.

Author response: I have made the required changes suggested by the reviewer.

Reviewer Comment: in Figure 3. FTIR of leaf capped AgNPs (A) and bulb capped AgNPs (B) extract of A. vittata, XRD 479 analysis of Bulb (C) and Leaf synthesized AgNPs (D) using A. vittata. ( give full length of FTIR spectra with zoom image.

Author response: We have provided the FTIR spectra as suggested by the reviewer.

Reviewer Comment: For antibacterial assay cite the following article Composite Anion Exchange Membranes with Antibacterial Properties for Desalination and Fluoride Ion Removal." ACS ES&T Water 1, no. 10 (2021): 2206-2216

Author response: we have incorporated the reference in the suggested activity.

Reviewer 2 Report

Overview and general recommendation:

The manuscript entitled "Biological Synthesis of Silver Nanoparticles by Amaryllis Vit- tata (L.) Herit: From Antimicrobial to Biomedical Applications " by Asad et al. reported the synthesis of silver nanoparticle (AgNP) bAmaryllis vittata 20 (L.) leaf and bulb extracts in greenway and its biological activity for human health benefits. It is a well-elucidated manuscript for the preparation of AgNP in a green and simple way and its biological activity. It may be accepted after considering the following points.

 Comments are below

  1. The author should mention how long time is required for AgNP synthesis by bulb and leaf extract at least through a change of color.
  2. Could the authors describe which solution, between bulb and leaf, gives the produced AgNP better stability?
  3. Figure 2: Justification is required as to why the beginning wavelength differs for the two extract systems. In this aspect, it has to be modified.
  4. Line 161: what is carbon coated SEM grid? Please explain it.
  5. Line 498: van der Walls forces not Vandar Waals.
  6. Authors should report the reproducibility of the biological activities of the AgNP.

There are several spelling errors and the writing style is highly informal. During the revision, kindly take a close look.

Author Response

Reviewer 2nd

Comments and Suggestions for Authors

Overview and general recommendation:

The manuscript entitled "Biological Synthesis of Silver Nanoparticles by Amaryllis Vit- tata (L.) Herit: From Antimicrobial to Biomedical Applications " by Asad et al. reported the synthesis of silver nanoparticles (AgNPs) by Amaryllis vittata 20 (L.) leaf and bulb extracts in greenway and its biological activity for human health benefits. It is a well-elucidated manuscript for the preparation of AgNPs in a green and simple way and its biological activity. It may be accepted after considering the following points.

Author Response: Worthy reviewer thanks for appreciating our work and giving the suggestion that improves our manuscript.

 Comments are below

Reviewer Comment: The author should mention how long time is required for AgNP synthesis by bulb and leaf extract at least through a change of color.

Author Response: Dear reviewer the color change for both types of AgNPs was observed after 10 minutes of reaction. The said answer is also reported in the paper.

Reviewer Comment: Could the authors describe which solution, between bulb and leaf, gives the produced AgNPs better stability?

Author response: The leaf-capped AgNPs were stable even after a month while the bulb-capped AgNPs were stable for up to 24 hours or up to 1 day.after that these AgNPs agglomerate.

Reviewer Comment: Figure 2: Justification is required as to why the beginning wavelength differs for the two extract systems. In this aspect, it has to be modified.

Author response: Dear reviewer we have corrected the figures. Now the beginning wavelengths are the same for both the leaf and bulb capped nps.

Reviewer Comment: Line 161: what is a carbon-coated SEM grid? Please explain it.

Author response Carbon coatings for electron microscopy are amorphous, conductive layers transparent to electrons. This implies that carbon coatings are particularly valuable for making non-conductive samples amenable to energy-dispersive x-ray spectroscopy (EDS)

Comment: Line 498: van der Walls forces not Vandar Waals.

Author response This mistake has been removed.

Reviewer Comment: Authors should report the reproducibility of the biological activities of the AgNPs.

Author response:The author has used the AgNP of A.vittata for biological activities. It was used for the first time to explore its biological potential as every plant has its medicinal properties. It can be reproducible as these were performed in triplicates.

There are several spelling errors and the writing style is highly informal. During the revision, kindly take a close look.

Author response The spelling and typographical mistakes were removed and the writing style is made formal. The manuscript was revised and checked.

Reviewer 3 Report

The manuscript is a mixture of every experiment authors could perform. Moreover, some of them are completely meaningless. The toxicity of silver NPs is not evaluated. However, the authors suggest such metal be used as a pain killer.  It should be rejected since the authors made crucial mistakes, and did not present XRD results, UV-Vis was given instead of FT-IR, and so forth. English presentation, as well as the presentation of the manuscript, is very poor. The specific comments are given in the pdf attached. 

Author Response

Reviewer 3rd

Comments and Suggestions for Authors

The manuscript is a mixture of every experiment authors could perform. Moreover, some of them are completely meaningless. The toxicity of silver NPs is not evaluated. However, the authors suggest such metal be used as a pain killer.  It should be rejected since the authors made crucial mistakes, and did not present XRD results, UV-Vis was given instead of FT-IR, and so forth. English presentation, as well as the presentation of the manuscript, is very poor. The specific comments are given in the pdf attached. 

Ans . Dear reviewer we have incorporated all the comments suggested by you in the attached pdf file. Silver capping by the plant reduces its toxicity and used to relieve pain in the said experiment. Also we have studied preliminary applications of these plant capped AgNPs. That’s why we have studied only few activities. FT-IR and XRD were missing because it may be due to some virus in the file.  I have again provided both FT-IR and XRD spectra. kindly have a look at these two figures.Also I have provided the graphical abstract.  The cytotoxicity of silver was not calculated in these preliminary applications. However further experiments can be done on these nanoparticles to calculate their cytotoxic, anticancer and other applications.

  1. In antibacterial activity the zone of inhibition are completely weird……?

Ans. We have provided some clear pictures and the zone of inhibition can be easily calculated by using these clear picture.

Round 2

Reviewer 1 Report

The author reviewed all suggested points 

Author Response

Thank you for considering and realizing our work. Your valuable suggestions and recommendations for the improvement of the article are highly appreciated.

Reviewer 3 Report

I recommend the manuscript "
Biological Synthesis of Silver Nanoparticles by Amaryllis Vittata (L.) Herit: From Antimicrobial to Biomedical Applications" for publication in Materials journal.
I reviewed this article some time ago and suggested rejection. Now, I see that the authors considered my suggestions, and other reviewers' comments were also considered. The current version of the manuscript seems very well.
In my opinion, the manuscript is ready for publication.

Author Response

(The authors gave the same response as above.)
